# Characterizing the cognitive and mental health benefits of exercise and video game playing

Sydni G. Paleczny[1☉]*, Conor J. Wild[1,2☉], Alex Xue[2], Roger Highfield[3], Adrian M. Owen[1,2,4]

1 Western Institute for Neuroscience, Western University, London, Ontario, Canada, 2 Department of Physiology and Pharmacology, Western University, London, Ontario, Canada, 3 Science Museum Group, London, United Kingdom, 4 Department of Psychology, Western University, London, Ontario, Canada

☉ These authors contributed equally to this work.
* spaleczn@uwo.ca

## Abstract

Two of the most actively studied modifiable lifestyle factors, exercise and video gaming, are regularly touted as easy and effective ways to enhance brain function and/or protect it from age-related decline. However, some critical lingering questions and methodological inconsistencies leave it unclear what aspects of brain health are affected by exercise and video gaming. In this cross-sectional global online study, we recruited over 1000 people and collected data about participants' physical activity levels, time spent playing video games, mental health, and cognitive performance using tests of short-term memory, verbal abilities, and reasoning skills from the Creyos battery. The amount of regular physical activity was not significantly related to any measure of cognitive performance; however, more physical activity was associated with better mental health as indexed using the Patient Health Questionnaire (PHQ-2) and Generalized Anxiety Disorder (GAD-2) screeners for depression and anxiety. Conversely, we found that more time spent playing video games was associated with better cognitive performance but was unrelated to mental health. We conclude that exercise and video gaming have differential effects on the brain, which may help individuals tailor their lifestyle choices to promote mental and cognitive health, respectively, across the lifespan.

## Introduction

The past decade has seen a proliferation of research into preventative medicine, and whether modifiable lifestyle factors can improve cognitive and mental health [1,2]. Two of the most well studied lifestyle factors, physical exercise and video game playing, are commonly suggested as activities that enhance cognitive functioning—such as the ability to store and recall information in memory, solve problems, and communicate—and protect these faculties from age-related decline [1–7]. It has also been suggested that physical activity and/or mental exercise in the form of video gaming

**Data availability statement:** All data files are held in a public repository, Borealis, The Canadian DataVerse repository: https://doi.org/10.5683/SP3/WUYAGU. Reference: Wild, Conor; Xue, Alex; Paleczny, Sydni; Highfield, Roger; Owen, Adrian, 2024, "The Brain & Body Study - complete raw dataset".

**Funding:** Funding was provided by the Canadian Institutes of Health Research (CIHR) program (grant awarded to A.M.O.).

**Competing interests:** The cognitive tests used in this study are marketed by Creyos (formerly Cambridge Brain Sciences), of which A.M.O. is the chief scientific officer, C.J.W. is the director of data science, and S.G.P. is staff scientist. Under the terms of the existing licensing agreement, A.M.O. and his collaborators are free to use the platform at no cost for their scientific studies, and that such research projects neither contribute to, nor are influenced by, the activities of the company. Consequently, there is no overlap between the present study and the activities of Creyos, nor was there any cost to the authors, funding bodies, or participants who were involved in the study. This does not alter our adherence to PLOS ONE policies on sharing data and materials.".

may positively influence mental health and well-being by decreasing symptoms of depression and anxiety [8–11]. While both activities may promote improved brain health, it isn't clear whether their benefits are similar; for example, do different activities enhance some cognitive domains more than others, or improve different aspects of mental health? Clearly, there is a need to better define the beneficial impacts of these particular lifestyle factors on brain health to both inform health policy and guide individuals toward interventions that truly promote healthy aging.

Many studies have suggested that a relationship exists between exercise, cognition, and mental health, although the exact nature of this relationship remains unclear [11–18]. Nevertheless, this literature has led the World Health Organization (WHO) to state that regular moderate-to-vigorous physical activity (MVPA) has "beneficial effects on cognition" for adults, older adults, and adults living with disability [19]. However, recent studies and meta-analyses suggest that the evidence supporting a causal link between exercise and cognition is tenuous; for example, low statistical power, small effect sizes, publication bias, mixed findings, and varying analytical designs cast significant doubt on the results of many studies [20–22]. Some researchers disagree with this position, and maintain that exercise yields direct benefits to cognition when key moderators such as age, health status, sex differences, as well as different types, intensities, and durations of physical activity are accounted for [23]. It is also worth noting that the precise cognitive functions studied within this research context have often been poorly defined and/or inconsistent. Some studies have described benefits to executive functioning, attention, or memory [13,24] although often these claims are made based on measures of global cognition pooled across dissimilar studies and inconsistent cognitive assessment tools [25–27]. Studies investigating the relationship between exercise and mental health have similar limitations, including publication bias, small effect sizes, and inconsistent results [8]. In short, it remains unclear exactly what aspects of cognitive function and psychological well-being are improved by exercise.

A similarly incomplete picture emerges from research seeking to establish a link between video game playing and brain function. For example, while it is well established that, in healthy adults, simplistic "brain training" games do not have any measurable cognitive impact beyond the specific task being trained [28–30], more complex, immersive, and demanding video games [31] may augment perceptual and cognitive abilities with extensive training [32–38]. Most of these positive findings, however, are based on studies of healthy young adults and may not generalize to older or patient populations. In fact, meta-analyses of the association between video gaming and brain health in older adults reveal conflicting results [5,6,39–41]. For example, some studies have suggested that video gaming may be beneficial to executive control processes including working memory, response inhibition and reasoning [42] as well as affective well-being [9], while other studies have shown it to be detrimental to aspects of cognitive and mental health [43–45].

To address some of the outstanding questions in this field, we performed an international online study in a large sample of younger, middle-aged, and older adults, with the objective of better characterizing the relationships between physical activity,

video game playing, cognition, and mental health. To measure various aspects of lifestyle and brain health, along with demographics and other health-related information, we deployed multiple validated questionnaires alongside a comprehensive cognitive test battery. The Creyos online cognitive testing platform, which includes twelve tasks that collectively assess a broad range of cognitive abilities, has been used in numerous large cohort studies [46–48] and provides detailed information about a variety of brain functions and the cognitive domains that they support. Based on the existing literature, we hypothesized that both exercise and video gaming would be related to cognition and/or mental health, although the exact nature of this relationship would likely vary in terms of the benefits observed – for example, the cognitive domains that showed associations with these lifestyle factors – and when during the lifespan they occur. To our knowledge, simultaneously assessing the effect of physical activity and video gaming on both cognitive performance and mental health in such a large sample has not been done previously. An advantage of this approach is that the profiles of brain health associated with each lifestyle factor can be directly compared because they are measured using the same instruments (i.e., questionnaires and cognitive tasks). The results from this large scale cross-sectional study will set the stage for future targeted studies of interventions and causal effects that will be of significant benefit for shaping health policy, informing individual lifestyle choices, and guiding future research on interventional approaches to brain health and longevity.

## Materials and methods

### Participants and study design

Recruitment was accomplished through online social media advertisements, radio and news broadcasts, and word of mouth between January 11, 2024 and October 8, 2024. Participants were eligible to participate if they had access to a laptop or desktop computing device with an active internet connection, were over the age of 18, able to read English, and did not have any motor difficulties. The study was approved by Western University's Research Ethics Board (protocol ID #124031, approved Dec 11, 2023) and all participants provided informed implied consent prior to participating. Participants were assigned unique alphanumeric study identifiers upon consenting to the study, which was stored alongside participant email address (used for study enrolment) in a master list available only to the Principal Investigator of the study. After reviewing the study procedures, participants were asked to complete an online demographic and health questionnaire, followed by the 12-task Creyos cognitive battery, then additional online questionnaires about physical activity and video gaming (see below).

### Measures

**Demographic and health questionnaires.**  Demographic and general health information included items about age, level of education, employment status, sex (as assigned at birth), gender (*"What gender do you identify as?"*, as free-form input), and diagnosed neurological, developmental, and pre-existing health conditions. Gender responses were inspected by the researchers and grouped into one of four emergent categories: male, female, non-binary, and other. In cases where the participant did not provide a gender or made an impertinent response, it was assumed that their gender identity matched their response to "sex assigned at birth" (if indicated).

**Physical activity.**  Participants completed the Physical Activity Adult Questionnaire (PAAQ; [49,50]) – which produces a self-reported estimate of the number of minutes of moderate-to-vigorous physical activity (MVPA) in multiple domains and in total for the past week. From the total number of MVPA minutes, participants were classified as to whether or not they met the WHO guidelines for physical activity (i.e., whether they achieved 150 minutes or more of MVPA in the past week).

**Video game playing.**  Video game use was quantified with the Bavelier Lab Video Game Questionnaire (VGQ; *Bavelier Lab Video Game Questionnaire*, 2022) version 2022 (adapted from Green et al., 2017). Briefly, participants were asked to estimate the average number of hours per week they spent playing different genres of video games within the past 12 months. The original scoring scheme for this instrument produces a very fine-grained categorization of gamers

who play action-based video games as it has strict criteria that delineate four types of "action video gamer players". In our sample, few participants met any of these criteria (i.e., most were not classified as action gamers), and most of those who played action games at all *also* played other genres. Given the strong overlap in gaming genres within our sample we would not have been able to separate genre-specific effects, so we simply categorized participants as non-gamers (zero hours per week), infrequent gamers (more than zero but less than three hours per week), or frequent gamers (3 + hours per week in one or more genres; a cut-point determined from the midpoint of the VGQ ordinal scale [37]). Although self-report data may differ from objective measurements due to inaccurate participant judgements [51], previous studies have demonstrated that a categorization method may address this problem and serve as a more appropriate use of self-report participant data as compared to a continuous time-based measure [37,52,53]. Previous studies have shown that, where self-report data is concerned, the categorization of individuals into extremes based on their reported quantity or frequency of the behaviour (e.g., whether or not they met activity guidelines, based on self-reported minutes) is more reliable than the continuous reported measure [37,49,53]. For example, in their original publication of the PAAQ, Colley et al. show that despite weak correlations (best case Pearson r = 0.36) between self-reported physical activity and objective measurements from accelerometers, the proportion of participants that met physical activity guidelines according to these methods were somewhat similar (39% vs. 46%) and these classifications were in agreement 67% of the time. Similarly, Bediou et al. [33] note that the video game questionnaire used in this study was designed only to distinguish frequent gamers from those who partake very infrequently.

**Measures of cognitive performance.** The 12-task Creyos battery assesses aspects of memory, attention, planning, and reasoning, using a web-based interface that can be self-administered at home through any internet browser [46]. The Creyos battery has been used in similar large-scale studies to assess the effects of sleep [47], COVID-19 [48], and lifestyle factors on cognitive functioning among the general population [54] across the lifespan [55] in studies with children [56,57] and older adults [54,58]. Importantly, several studies have demonstrated that older adults with limited experience with technology can manage digital cognitive assessments, producing data that is as reliable as in-person assessment [59–61]. Furthermore, these tests have been validated in patients with anatomically specific brain lesions, in neurodegenerative populations, in pharmacological intervention studies, and their neural correlates have been well studied using functional neuroimaging in healthy adults [46]. The 12 tests (detailed in the supplementary materials of some of our previous studies; e.g., Hampshire et al., 2012; Wild et al., 2018) are: 1) Number Ladder (visuospatial working memory); 2) Spatial Span (spatial short-term memory); 3) Token Search (working memory); 4) Paired Associates (episodic memory); 5) Rotations (mental rotation); 6) Grammatical Reasoning (verbal reasoning); 7) Polygons (visuospatial processing); 8) Odd One Out (deductive reasoning); 9) Spatial Planning (planning); 10) Feature Match (attention); 11) Double Trouble (response inhibition); 12) Digit Span (verbal short-term memory).

Rather than separately examining the individual outcomes of the cognitive tasks, which have been shown to share considerable variance [46,47], we used five higher-level composite scores of cognitive performance as the dependent variables in our analyses. These included three measures derived from a factor analysis [46,47] that reflect performance within three broad cognitive domains: 1) short-term memory (STM; the ability to focus on and maintain task-relevant information in short-term memory), 2) reasoning (the ability to mentally transform information according to logical rules), and 3) verbal performance (the processing of information within the verbal or linguistic domain). These interpretations are derived from the characteristics shared by the tasks that load most heavily on each of the three factors observed by Hampshire et al. (2012), and are supported by neuroimaging results showing that distinct (and functionally consistent) brain networks are recruited by these combinations of tasks. A fourth measure, processing speed, was derived from reaction-time-based markers from the cognitive tasks [48] and a fifth measure was calculated as the overall (average) performance when all 12 individual tasks were considered. These five composite scores were calculated using transformations and parameters that were estimated using an entirely different data set – the normative group (N = 7,832) described in Wild et al., 2022; note the factor solution for the composite scores was nearly identical in the normative sample and the current study group,

as Tucker's congruency coefficient was greater than 0.95 for all domains, indicating that the solutions could be considered equal [62–64].

First, individual test scores were centered (mean = 0) and power transformed (Yeo-Johnson transformed) to have a standard deviation of 1.0. The three domain scores were calculated using the regression method (i.e., multiplying the 12 standardized scores by pseudo-inverse of the factor loading matrix and summing across tasks), where the loading matrix was derived from a Varimax-rotated Principal Components Analysis (PCA) of the normative dataset. The processing speed score was calculated similarly, except using standardized reaction-time based measures from the tasks and the loadings from only the first principal component of these measures in the normative group. The overall score was calculated as the average of the 12 standardized test scores. Finally, all composite scores were rescaled to have mean of zero and standard deviation of one within this study sample.

**Measures of mental health.** Mental health was assessed using the Generalized Anxiety Disorder (GAD-2) screener for anxiety symptoms [65] and the Patient Health Questionnaire (PHQ-2) screener for symptoms of depression [66]. The GAD-2 screener consists of two items based on the GAD-7 [67] and has demonstrated acceptable sensitivity and specificity for categorizing generalized anxiety disorder [68]. The PHQ-2 screener consists of the first two items of the PHQ-9 [69], which ask about the frequency of depressed mood and anhedonia, and has been shown to have sensitivity of 83% and specificity of 92% for categorizing major depression [66]. For both scales, ordinal scores are obtained by summing the points from each question, with a total score of 3 or greater marking the cut-off for identifying possible generalized anxiety disorder (GAD-2) or depression (PHQ-2).

## Data analysis

**Software.** Most analyses were performed in Python (version 3.9.12), using the following packages: Datalad (v1.0.0; data history and tracking), Pandas (v2.2.2; for data preprocessing and manipulation), Numpy (v1.2.4; numeric computing backend), Statsmodels (v0.13.5; building and estimating linear and ordinal regression models).

Additional analyses were performed in R (v4.4.1) with the packages MASS (v7.30-60.2; ordinal regression models), brant (v0.3-0; Brant test of the parallel regression assumption for ordinal logit models), emmeans (v1.10.4; estimating and displaying marginal means and probabilities from regression models), ggplot2 (v3.5.1; plots), and dplyr (v1.1.4) and tidyverse (v.2.0.0) for data manipulation.

All code used for cleaning, running analyses, and creating figures is available at: https://osf.io/vng9a/.

**Data preprocessing.** Multiple datasets (three questionnaires and one set of cognitive test scores) were exported from their respective platforms and preprocessed to convert data field types, remove extraneous fields, and score questionnaire materials, and then were combined using participants' unique study identifiers. The combined dataset was then subjected to a series of filtering stages to remove participants: with invalid ages, who did not speak English, who reported a diagnosis of Alzheimer's disease, other forms of dementia, Parkinson's disease, multiple sclerosis, a neurodevelopmental motor disorder, or cancer. Note, individuals with the diagnoses listed above were excluded from our analyses because it was not known whether their diagnosis was current, whether the disease directly affected brain function (e.g., brain tumors), and/ or if they were receiving treatment(s) that might affect cognition. Those who reported unusually high numbers for numeric entry questions (according to visual inspection of histogram data to identify outliers), who had a cognitive score more than 4 standard deviations from the mean, or who were missing any data from the cognitive tests, mental health screeners, the PAAQ, or the VGQ were also excluded. Thresholds for "unusually high" numbers were determined using visual inspection of discontinuities and/or breakpoints within the data for each question. See Table 1 for details and results.

**Statistical analyses.** Linear multiple regression was used to predict each of the five measures of cognitive performance from participants' lifestyle factors (physical activity and video gaming) and additional covariates. In these five models, the effect of physical activity was modeled as a single binary predictor (i.e., "passed_WHO_guidelines"; whether or not participants met the WHO guidelines for at least 150 minutes of MVPA during the past week) and the effect of video

**Table 1. Data Filtering Stages.**

| Preprocessing Stage | Total N | Delta |
|---|---|---|
| Merge datasets | 1412 | +1412 |
| Age outside of range 18–98 years | 1404 | −8 |
| Does not speak English | 1337 | −67 |
| Diagnosed with neurological condition or cancer | 1292 | −45 |
| Reported >10 concussions | 1285 | −7 |
| Consumes >30 units of caffeine/ week. | 1276 | −9 |
| Reported consuming >20 units of alcohol/ day | 1185 | −91 |
| Reported consuming >20 units of cannabis/ day | 1179 | −6 |
| Reported smoking >20 cigarettes/ day | 1169 | −10 |
| Speaks >7 languages | 1152 | −17 |
| PAAQ >1800 minutes of MVPA/ week | 1028 | −124 |
| At least one cognitive test score >4 SDs from mean | 1003 | −25 |
| At least one cognitive test score missing | 984 | −19 |
| GAD-2/ PHQ-2 – no data available | 979 | −5 |
| PAAQ – no data available | 973 | −6 |
| VGQ – no data available | 923 | −50 |

Remaining total sample size (Total N) is reported following each step, in addition to the change in sample size from the previous step (Delta). Thresholds for "unusually high" numbers were determined using visual inspection of discontinuities and/or breakpoints within the data for each question.

gaming was modeled with a set of two dummy regressors that coded infrequent and frequent gamers (baseline being non-gamers). Covariates of no interest included: age (mean centered continuous linear regressor), gender (2 binary dummy predictors for female and other+nonbinary categories, with male as the baseline), completion of post-secondary education (binary predictor), and the presence of a pre-existing health condition (binary predictor indicating one or more of diabetes, obesity, hypertension, stroke, heart attack, coronary artery disease, or concussion). The parameters of these 5 models were estimated using ordinary least squares (OLS) because the dependent variables (and the residuals after model fitting) were normally distributed.

Given the non-normal outputs of the GAD-2 and PHQ-2, ordinal regression [70] was used to model the associations between these measures and physical activity, video gaming, and covariates; specifically, we used probit ordinal regression because this is the default ordinal regression model in the software used for our analyses. The regression models included all the same predictors as the OLS regression models with additional parameters for the thresholds between levels of the ordinal scale and were estimated using maximum likelihood.

A key assumption underlying ordinal regression is that of proportionality – the parallel regression or parallel lines assumption – which assumes that the effect of each predictor variable should be the same across all levels of the outcome variable. We tested this assumption for PHQ-2 and GAD-2 models using the Brant test [71], which estimates separate logistic regression models for each threshold of the outcome and compares the coefficients across levels. If the effect of a predictor is significantly different across thresholds, it indicates that the proportionality assumption is violated for that predictor. The test could not converge given the low number of samples with a gender of "other", so we excluded these data rows from any ordinal regression models. The test also produces an overall result that indicates whether any of the predictors violate the proportionality assumption. According to this test, the ordinal regression model for PHQ-2 scores did not violate this assumption ($\chi^2_{(35)} = 29.91$, $p > 0.05$), but there was evidence that the model for GAD-2 scores did ($\chi^2_{(35)} = 50.84$, $p < 0.05$). Therefore, we collapsed the scale into four new categories: 0−1, 2−3, 4−5, and 6. A regression model for this alternative GAD-2 scale satisfied the Brant test for proportionality ($\chi^2_{(14)} = 23.05$, $p > 0.05$).

To statistically test the relationship between a brain health measure (a dependent variable) and a lifestyle factor (a set of independent variables), likelihood ratio tests (LRT) were used to compare two nested models: the full model ($H_1$) containing all factors and covariates, and a reduced model ($H_0$) that did not contain the regressors belonging to the factor of interest. A significant LRT indicates that the more complex model, which included the additional predictors of interest, provided a better fit to the data; in other words, that there is a relationship between the additional factor and the dependent variable after accounting for all covariates and the other lifestyle factor. Results were considered statistically significant if $p < 0.05$ when Bonferroni-corrected for 14 comparisons (two factors – physical activity and video gaming – for each of seven models).

Following a significant LRT, we used estimated marginal means computed from the $H_1$ (full) model to compare the outcome score (adjusted for covariates) between levels of the lifestyle factor. Pairwise comparisons of the adjusted means were conducted to test for statistically significant differences, and $p$-values were Bonferroni adjusted for the number of comparisons within each set of tests.

## Results

### Participants

A total of 2627 individuals signed up to participate via our online study platform. 2513 participants remained after removing duplicate email addresses and invalid entries. Of those, 1412 appeared in the cognitive dataset and at least one questionnaire (Table 1 – first row). After filtering for unlikely responses, outliers, and missing data, the final sample retained 923 individuals (Table 1). Participant demographics and health characteristics are summarized in Table 2.

### Data descriptives

Descriptive statistics of lifestyle factors and mental health measures are shown in Table 2. On the GAD-2 and PHQ-2 screeners, the mode for each scale was a score of zero, which corresponds to good mental health. The number of participants who were flagged for potential generalized anxiety disorder (i.e., GAD-2 >= 3) was 63, and the number of participants flagged for depression (PHQ-2 >= 3) was 52.

Statistics describing the 12 raw cognitive test scores are reported in S1 Table. As described in the methods, composite cognitive scores were re-scaled to have mean of zero and SD of one.

### Associations between lifestyle factors, cognition, and mental health

**Physical activity.** Likelihood ratio tests were used to examine whether the effect of physical activity (i.e., > 150 minutes MVPA in the past week) was associated with cognitive scores or mental health screeners (see S2 Table for complete statistics of all comparisons). There were no significant associations between physical activity and cognitive scores (all uncorrected $p$-values > 0.05). However, there was a significant association with PHQ-2 scores ($LR_{(1)} = 12.37$, $p_{unc} < 0.05$, $p_{adj} < 0.05$) and GAD-2 scores ($LR_{(1)} = 9.29$, $p_{unc} < 0.05$, $p_{adj} < 0.05$). Parameters and associated statistics from full $H_1$ OLS regression models (cognitive scores) and ordinal regression models (mental health scores) are reported in Tables 3 and 4, respectively. For both mental health screeners, the "passed_WHO_guidelines" coefficients were negative (Table 4), indicating that meeting the WHO recommendation for physical activity was associated with lower (better) scores on both of these mental health screeners.

To better describe this association, we examined the estimated marginal probabilities (similar to estimated marginal means) of obtaining a specific screener score given the participants' physical activity (Fig 1). The statistical results of all pairwise comparisons between predicted marginal probabilities are presented in S3 Table. In summary, on the PHQ-2, individuals who met the WHO guidelines were significantly more likely to score zero (estimate = 0.12, $z = 3.50$, $p_{adj} < 0.05$), and less likely to score 1 (estimate = −0.02, $z = −3.33$, $p_{adj} < 0.05$) and 2 (estimate = −0.05, $z = −3.39$, $p_{adj} < 0.05$). The

**Table 2. Participant Demographics, Health Characteristics, Lifestyle Factors & Brain Health Measures.**

| Measure | Value |
|---|---|
| **n** | 923 |
| **Age,** mean (*SD;* IQR) | 55.2 (*14.4*; 47.0 - 65.0) |
| **Gender,** n (%) | |
| Female | 563 (61.0%) |
| Male | 351 (38.1%) |
| Non-binary | 6 (0.6%) |
| Other | 3 (0.3%) |
| **Education Level,** n (%) | |
| Completed postsecondary education | 769 (83.3%) |
| Less than (or some) postsecondary education | 154 (16.7%) |
| **Employment Status,** n (%) | |
| Retired | 310 (33.6%) |
| Not retired | 613 (66.4%) |
| **Country of Residence (Top 4),** n (%) | |
| United Kingdom | 713 (77.2%) |
| Canada | 123 (13.3%) |
| USA | 29 (3.1%) |
| France | 10 (1.1%) |
| **Pre-Existing Health Conditions,** n (%) | |
| Hypertension | 103 (11.2%) |
| Diabetes | 49 (5.3%) |
| Obesity | 48 (5.2%) |
| Coronary artery disease | 23 (2.5%) |
| Concussion | 22 (2.4%) |
| Stroke | 12 (1.3%) |
| Heart attack | 11 (1.2%) |
| **Neurological Diagnoses (Top 4),** n (%) | |
| ADHD | 23 (2.5%) |
| Dyslexia | 21 (2.2%) |
| ASD | 15 (1.6%) |
| Dyspraxia | 2 (0.2%) |
| **Physical Activity** | |
| PAAQ MVPA mins/week, mean (*SD*, min – max) | 356.2 (*291.3*, 0.0 - 1740.0) |
| Passed WHO Guidelines, n (%) | 678 (73.2%) |
| Did not pass WHO Guidelines, n (%) | 247 (26.8%) |
| **Video Gaming,** n (%) | |
| Frequent gamers | 239 (24.8%) |
| Infrequent gamers | 378 (41.0%) |
| Non gamers | 316 (34.2%) |
| **PHQ-2 Score,** n (%) | |
| 0 | 628 (68.0%) |
| 1 | 124 (13.4%) |
| 2 | 119 (12.9%) |
| 3 | 19 (2.1%) |
| 4 | 12 (1.3%) |

*(Continued)*

**Table 2.** (Continued)

| Measure | Value |
|---|---|
| 5 | 8 (0.9%) |
| 6 | 13 (1.4%) |
| **GAD-2 Score,** n (%) | |
| 0 | 541 (58.6%) |
| 1 | 193 (20.9%) |
| 2 | 126 (13.7%) |
| 3 | 19 (2.1%) |
| 4 | 19 (2.1%) |
| 5 | 10 (1.1%) |
| 6 | 15 (1.6%) |

*Note.* On the PAAQ, 81.5% of respondents said that this was characteristic of their typical routine.

differences were diminished at the higher end of the scale, as there were no significant differences between predicted probabilities for PHQ-2 scores of 4, 5, or 6. A similar pattern was observed for the GAD-2: those who met the WHO guidelines for PA were more likely to score 0−1 (estimate = 0.10, z = 2.96, $p_{adj} < 0.05$) and less likely to score 2−3 (estimate = −0.06, z = −2.99, $p_{adj} < 0.05$).

**Video gaming.** The overall effect of video gaming (i.e., non-gamers, infrequent gamers, frequent gamers) on cognitive performance and mental health was again tested using likelihood ratio tests. After correcting for multiple comparisons, we found a significant association between video gaming and four of five cognitive performance scores: STM ($LR_{(2)}$ = 18.68, $p_{unc} < 0.05$, $p_{adj} < 0.05$), reasoning ($LR_{(2)}$ = 21.78, $p_{unc} < 0.05$, $p_{adj} < 0.05$), processing speed ($LR_{(2)}$ = 24.06, $p_{unc} < 0.05$, $p_{adj} < 0.05$), and overall cognitive score ($LR_{(2)}$ = 41.61, $p_{unc} < 0.05$, $p_{adj} < 0.05$). There were no significant associations between video gaming and PHQ-2 or GAD-2 scores (all corrected $ps > 0.05$; S2 Table). Regression parameters and associated statistics from OLS regression models (cognitive scores) and ordinal regression models (mental health scores) are reported in Tables 3 and 4, respectively.

Next, we used estimated marginal means and tests of pairwise differences to describe the associations between cognitive scores and playing video games. Fig 2 displays the estimated marginal means of each cognitive score for each type of video gamer and summarizes the patterns of statistical differences between these groups. S4 Table contains statistics for all the pairwise comparisons of estimated marginal means within each cognitive score. Simply, for the measures of overall performance and processing speed, frequent gamers performed better than infrequent gamers (adjusted $ps < 0.05$) and infrequent gamers performed better than non-gamers (adjusted $ps < 0.05$). In the STM domain, frequent and infrequent gamers scored significantly better than non-gamers (adjusted $ps < 0.05$) but there was no difference between those who played video games. In the reasoning domain, frequent gamers performed better than infrequent gamers and non-gamers (adjusted $ps < 0.05$) but there was no difference between non-gamers and infrequent gamers.

**Age, brain health, and lifestyle factors.** It has been proposed that physical activity and video gaming might be more important to brain health as we get older. In other words, the benefits afforded by these interventions might especially impact (or only be observed in) older individuals. To explicitly test this hypothesis, we examined whether there was a significant interaction between participants' age and the lifestyle factors when predicting cognitive scores in adults older than 50 years. As in the previous analyses, we used LRTs to test whether the inclusion of the interaction term (age*passed_WHO_guidelines, age*gamer_type) improved the model fit. However, we found no significant associations, even at an uncorrected level (S5 Table). To illustrate an example of this, we plotted the estimated marginal means for cognitive scores, as a function of age and physical activity status (Fig 3); these plots clearly show that the downward trends of these scores with increasing age do not vary with physical activity.

**Table 3. OLS parameter estimates.**

| Regression Parameter | Score | Coef | SE | t | df | p.unc | 95% CI lower | 95% CI upper |
|---|---|---|---|---|---|---|---|---|
| passed_who_guidelines | STM | −0.11 | 0.07 | −1.62 | 914 | 0.105 | −0.25 | 0.02 |
| | Reasoning | 0.13 | 0.07 | 1.96 | 914 | 0.050 | 0.00 | 0.26 |
| | Verbal | 0.01 | 0.07 | 0.12 | 914 | 0.902 | −0.14 | 0.15 |
| | Processing Speed | 0.06 | 0.06 | 0.99 | 914 | 0.322 | −0.06 | 0.18 |
| | Overall | 0.02 | 0.06 | 0.34 | 914 | 0.732 | −0.10 | 0.14 |
| frequent_gamer | STM | 0.36 | 0.09 | 4.23 | 914 | < 0.001 | 0.19 | 0.53 |
| | Reasoning | 0.39 | 0.08 | 4.66 | 914 | < 0.001 | 0.22 | 0.55 |
| | Verbal | 0.06 | 0.09 | 0.66 | 914 | 0.510 | −0.12 | 0.24 |
| | Processing Speed | 0.37 | 0.08 | 4.87 | 914 | < 0.001 | 0.22 | 0.52 |
| | Overall | 0.49 | 0.08 | 6.49 | 914 | < 0.001 | 0.34 | 0.63 |
| infrequent_gamer | STM | 0.20 | 0.08 | 2.87 | 914 | 0.004 | 0.06 | 0.34 |
| | Reasoning | 0.14 | 0.07 | 2.01 | 914 | 0.044 | 0.00 | 0.27 |
| | Verbal | −0.07 | 0.08 | −0.89 | 914 | 0.376 | −0.22 | 0.08 |
| | Processing Speed | 0.19 | 0.06 | 2.99 | 914 | 0.003 | 0.06 | 0.31 |
| | Overall | 0.19 | 0.06 | 3.04 | 914 | 0.002 | 0.07 | 0.31 |

OLS parameter estimates for effects of interest from the full (H1) regression models estimated for each cognitive score. Additional covariates (not shown) included: age, gender, education, pre-existing health condition. STM = short-term memory, Coef = coefficient value, SE = standard error, t = t-statistic, df = degrees of freedom, p.unc = uncorrected p-value for the associated t-statistic, CI = 95% confidence intervals of the parameter estimate.

**Table 4. Ordinal regression parameters for effects of interest.**

| Regression Parameter | Score | Coef | SE | t | df | p.unc | 95% CI Lower | 95% CI Upper |
|---|---|---|---|---|---|---|---|---|
| passed_who_guidelines | phq2 | −0.32 | 0.09 | −3.53 | 901 | < 0.001 | −0.50 | −0.14 |
| | gad2 | −0.32 | 0.10 | −3.07 | 904 | 0.002 | −0.52 | −0.11 |
| infrequent_gamer | phq2 | −0.03 | 0.10 | −0.28 | 901 | 0.781 | −0.23 | 0.17 |
| | gad2 | −0.04 | 0.11 | −0.38 | 904 | 0.704 | −0.26 | 0.18 |
| frequent_gamer | phq2 | 0.24 | 0.12 | 2.12 | 901 | 0.034 | 0.02 | 0.47 |
| | gad2 | −0.01 | 0.13 | −0.09 | 904 | 0.929 | −0.27 | 0.25 |
| 0.0/1.0 | phq2 | −0.89 | 0.23 | −3.83 | 901 | < 0.001 | −1.34 | −0.43 |
| | gad2 | −1.05 | 0.26 | −4.04 | 904 | < 0.001 | −1.57 | −0.54 |
| 1.0/2.0 | phq2 | −0.79 | 0.08 | −9.03 | 901 | < 0.001 | −0.95 | −0.62 |
| | gad2 | −0.07 | 0.08 | −0.88 | 904 | 0.379 | −0.22 | 0.08 |
| 2.0/3.0 | phq2 | −0.28 | 0.09 | −3.26 | 901 | 0.001 | −0.45 | −0.11 |
| | gad2 | −0.59 | 0.18 | −3.27 | 904 | 0.001 | −0.94 | −0.24 |
| 3.0/4.0 | phq2 | −1.43 | 0.22 | −6.42 | 901 | < 0.001 | −1.86 | −0.99 |
| 4.0/5.0 | phq2 | −1.51 | 0.28 | −5.36 | 901 | < 0.001 | −2.06 | −0.96 |
| 5.0/6.0 | phq2 | −1.51 | 0.35 | −4.39 | 901 | < 0.001 | −2.19 | −0.84 |

Ordinal regression parameters (estimated with maximum likelihood) for the effects of interest from regression models estimated for each mental health score. Additional covariates (not shown) included: age, gender, education, pre-existing health condition. Coef = coefficient value, SE = standard error, t = t-statistic, df = degrees of freedom, p.unc = uncorrected p-value for the associated t-statistic, CI = 95% confidence intervals of the parameter estimate.

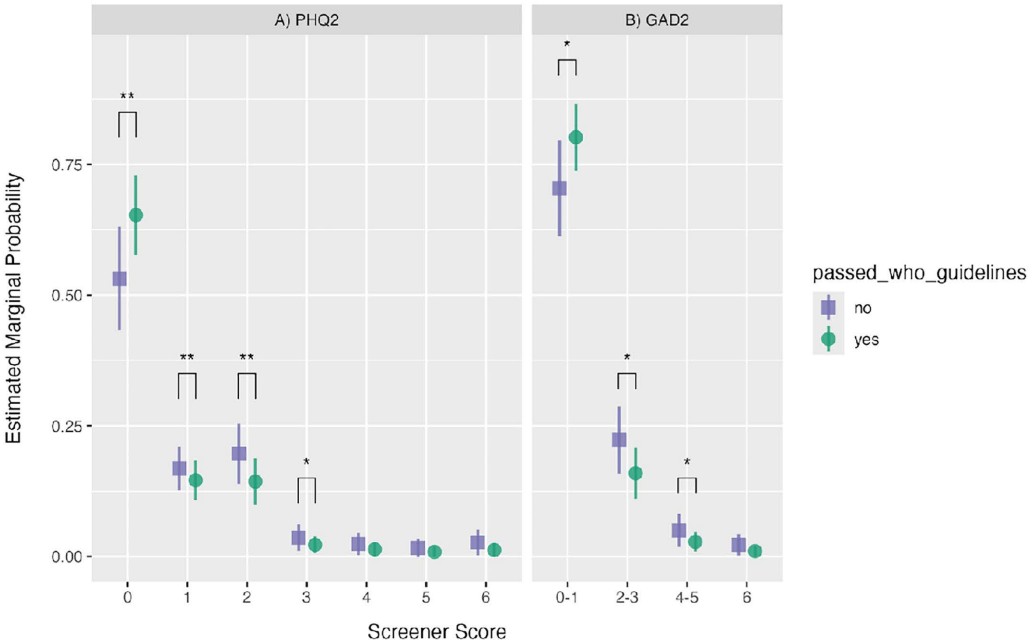

**Fig 1. Estimated Marginal Probabilities of Mental Health Screener Scores Given Exercise Level.** Estimated marginal probabilities of possible scores on the **A)** PHQ-2 and **B)** GAD-2, given the level of physical activity ("passed_WHO_guidelines"). Data colour and shape indicate the level of the physical activity predictor (purple square = did not meet WHO MVPA guidelines; green circle = met or exceeded the WHO MVPA guidelines). Note that the GAD-2 was collapsed into fewer levels (4 instead of 7) to satisfy statistical assumptions for ordinal regression. Error bars represent 95% confidence intervals of the predicted probability. Brackets and asterisks indicate significant pairwise differences between groups at each possible score, Bonferroni-corrected for **A)** N = 7 and **B)** N = 4 comparisons (* $p_{adj}$ < 0.05; ** $p_{adj}$ < 0.01; *** $p_{adj}$ < 0.001).

## Discussion

From an initial international sample of more than 1000 participants, we demonstrated that physical activity and video gaming have different effects on cognitive performance and mental health. Physical activity was not associated with any measure of cognition; however, more frequent physical activity was associated with better mental health. In comparison, playing video games was found to be associated with better cognitive performance, but was not associated with mental health. It is important to clarify that the results do not indicate that one intervention is necessarily better than another; rather, that video gaming and physical exercise appear to have positive effects on cognition on the one hand, and mental health on the other.

### Physical activity and brain health

The present findings do not dispute the established benefits of physical activity on physiological health. That is to say, physical exercise undoubtedly improves physical fitness and has well-documented beneficial effects for general health [72]. However, previous studies have also reported that regular exercise is associated with improved executive function [13,24,73], a delay in onset of dementia [74], and better mental health [11,17]. These findings relate to a wide range of physical activities including aerobic exercise [75], resistance training [76], and general physical activity with varied durations and intensities [77]. In contrast, in the present study, we found no evidence that meeting the WHO's recommendation of physical activity, according to self-reported estimates, is related to cognitive ability. This result is consistent with recent meta-analyses and studies that have reported limited or no benefit of physical activity for cognition [20,22,25–27,78–80]. However, in the present study, we were unable to account for all potential exercise-related factors that may exert influence on cognition (e.g., the type of exercise performed, acute or chronic activity, etc.). We cannot definitively say that *no*

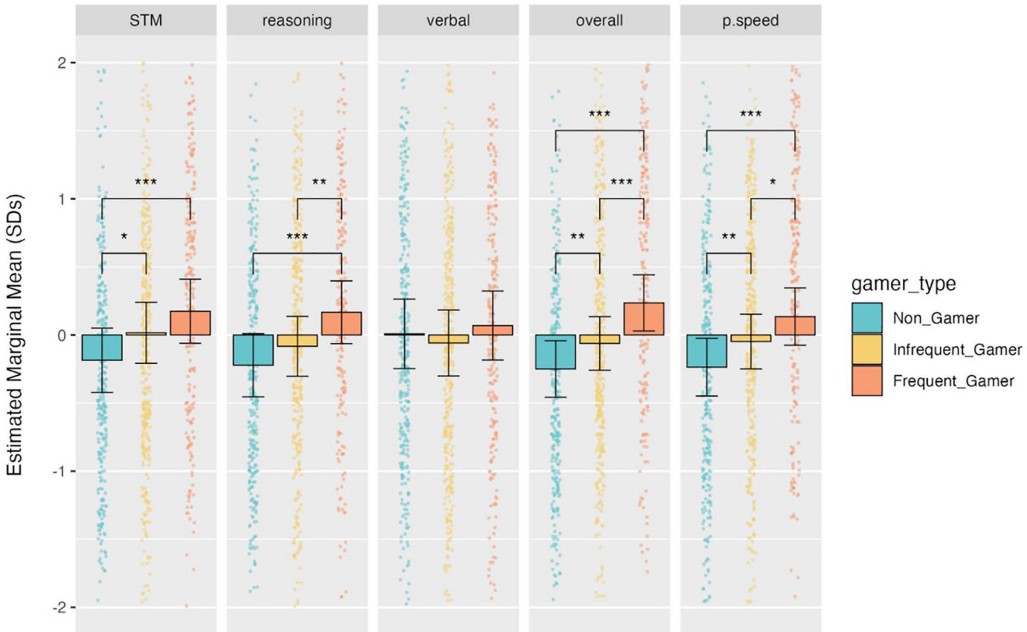

**Fig 2. Estimated Marginal Means of Cognitive Scores.** Coloured bars indicate the estimated marginal mean for each type of video game player (for each cognitive score) and error bars show the 95% confidence interval of the estimate. Individual raw data points are shown in the corresponding colour over each bar, but note that some data points are not displayed because the y-axis is truncated to +/- 2.0. Brackets and asterisks indicate significant pairwise differences between groups within each score type, Bonferroni-corrected for N = 3 comparisons (* $p_{adj}$ < 0.05; ** $p_{adj}$ < 0.01; *** $p_{adj}$ < 0.001).

relationship exists between alternate accounts of physical activity and cognition. Instead, it may be more appropriate to interpret the WHO guidelines as beneficial for physical health and well-being, as opposed to directly linking the guidelines with improved cognitive function. Ultimately, any recommendation of regular physical exercise need not rely on the *potential* for cognitive benefits, as the widespread health benefits of physical activity have been well documented and support this concept regardless [21].

Indeed, the current study also included an assessment of psychological health and well-being, and the results suggest that more frequent physical activity is associated with better mental health. Specifically, physical activity that exceeded WHO guidelines was associated with better scores on the PHQ-2 and the GAD-2 (reduced levels of depression and anxiety). In addition, there was a difference between those who had symptoms of anxiety/depression versus those who did not, based on whether they passed WHO guidelines. Specifically, on the PHQ-2, individuals who met the WHO guidelines were significantly more likely to score zero (no depressive symptoms), and less likely to score 1 or 2 (mild depressive symptoms). Similarly, on the GAD-2, those who met the WHO guidelines were more likely to score 0−1 (no symptoms of anxiety) and less likely to score 2−3 (mild anxiety). To put this result in perspective, the probability of scoring less than 2 on the PHQ-2 (better mental health) was approximately 80% when the WHO exercise recommendations were exceeded. But this probability was almost 10% less when the recommendations were *not* exceeded. Notably, differences were not observed at the higher end of either the PHQ-2 or the GAD-2, although reduced numbers of participants in those categories likely led to reduced statistical power. Nevertheless, this observation suggests that the effects observed were not driven by the severely depressed or anxious individuals in our sample; rather physical activity made the greatest difference at the lower end of both scales where symptoms were either mild or absent altogether.

These findings contribute to a growing body of evidence suggesting that regular exercise can significantly improve mental well-being. One of the primary physiological mechanisms driving this relationship is likely to be the modulation of

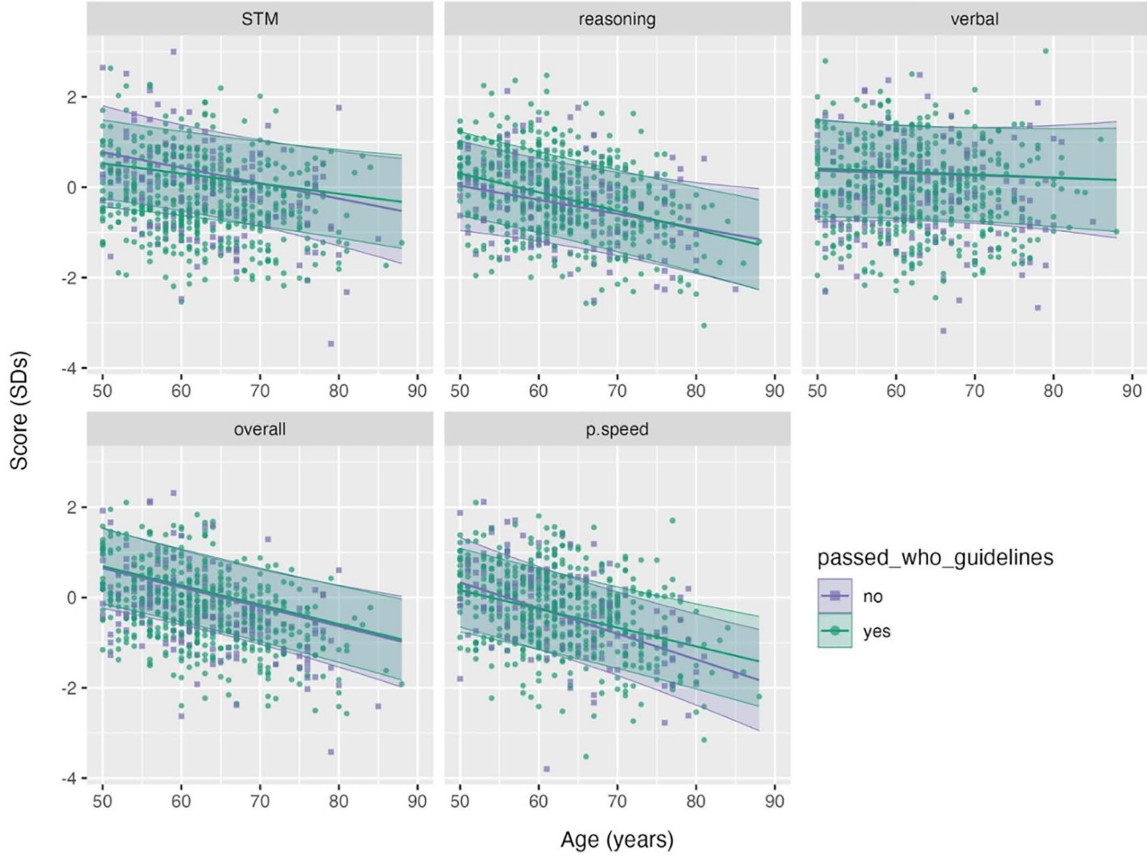

**Fig 3. Cognitive Scores as a Function of Age.** Cognitive scores as a function of age for participants older than 50. Lines and shaded regions indicate the estimated marginal mean (predicted) score across the plotted age range with 95% confidence intervals. Raw data are plotted as individual data points, where colour and shape correspond to the group: (purple/ circle) those that did not meet the WHO recommendations for 150 minutes of MVPA/ week, and (green/ triangle) those that did.

neurotransmitter systems, particularly those involved in mood regulation [81]. For example, physical activity stimulates the release of endogenous opioids, including endorphins, which are thought to produce feelings of euphoria, often referred to as the "runner's high" [82,83]. Regular physical activity is also believed to increase the synthesis and release of serotonin, dopamine, and norepinephrine, neurotransmitters that are crucial for mood regulation and are often targeted by antidepressant medications [84,85]. By upregulating these neurotransmitters, regular physical exercise may contribute to improved mood and reduce symptoms of depression and anxiety.

Another important physiological pathway is the hypothalamic-pituitary-adrenal axis, which is intimately linked to the body's stress response. Chronic stress is a well-established risk factor for mental health disorders, particularly depression and anxiety [86,87]. Exercise has been shown to regulate the hypothalamic-pituitary-adrenal axis by reducing baseline cortisol levels and enhancing the body's ability to cope with acute stress [88]. This stress-buffering effect may be another reason why the physically active individuals in our sample reported lower levels of depression and anxiety.

Physical activity is also associated with changes in brain-derived neurotrophic factor (BDNF), a protein that supports the growth, maintenance, and survival of neurons. Exercise has been shown to upregulate BDNF levels, leading to increased neurogenesis and synaptic plasticity, which may have positive effects on mental health [89,90].

Physical activity unquestionably improves cerebrovascular health [72], which increases blood flow to the brain, enhancing oxygenation and the delivery of nutrients. This improved vascular function can help protect against cognitive decline and enhance brain function [91], which may contribute to better mental health outcomes. Increased blood flow also helps to reduce inflammatory markers in the brain like C-reactive protein (CRP) and interleukin-6 (IL-6), which have been implicated in the pathophysiology of depression [92].

Finally, physical activity has been shown to improve sleep quality, which plays a critical role in mental health [93,94]. Exercise can help regulate circadian rhythms and reduce symptoms of insomnia [95,96], both of which are closely linked to emotional well-being. Better sleep contributes to improved mood [97], cognitive function [47], and overall mental health [94].

### Video gaming and brain health

In this study, video gaming was found to be associated with improvements in cognitive ability, but not mental health. As noted in the Introduction, the previous literature in this area lacks consensus. For example, some studies have suggested that video gaming may be beneficial to executive control processes including working memory, response inhibition and reasoning [42] as well as affective well-being [9], while other studies have shown it to be detrimental to aspects of cognitive and mental health [43–45]. The current study benefited from having a relatively large number of participants and a comprehensive suite of cognitive tasks that assessed aspects of reasoning, problem-solving, attention, working memory, visuospatial and verbal skills. Indeed, close inspection of the results presented in Fig 2 suggests that some aspects of cognitive function (e.g., short-term memory and reasoning) may have benefited more from frequent video gaming than others (e.g., verbal skills); though, we did not explicitly test differences between domains. This may in part explain some of the inconsistencies observed in previous studies; particularly those in which the precise cognitive functions studied have been poorly defined and/or inconsistent [39,41].

In the current study, self-report data was used to classify individuals as non-gamers, infrequent gamers or frequent gamers instead of relying on a continuous time-based measure. When comparing frequent gamers to non-frequent gamers, we observed similar results to when the total number of gaming hours was used. For example, there was a small difference between 0 and 0-3h, indicating a modest cognitive benefit for infrequent gamers compared to non-gamers. This suggests that the effects of video gaming on cognition not only depend on whether you play video games or not, but also on how often you play.

There are a variety of possible mechanisms by which video gaming may enhance cognition. For example, functional neuroimaging studies have shown that video games are associated with increased gray matter volume in the hippocampus [98,99], a region critical for memory and spatial reasoning [100]. Video games also appear to enhance the functioning of the prefrontal cortex, which is involved in higher-order cognitive processes like decision-making, attention, and task-switching [101,102]. Playing video games has also been found to increase the release of the neurotransmitter dopamine, presumably through its involvement in reward processing, motivation, and learning [103]. Whether these activities produce long-term changes in dopamine function is less clear, although in animal models, long-term exposure to stimulating environments has been shown to increase dopamine receptor density and improve dopamine-related behaviors, suggesting potential long-term benefits for humans [104].

In addition to these structural and chemical changes, video games are well known to enhance visual attention and processing speed, through repetition and practice [105,106]. Action gamers tend to exhibit improved visual acuity and faster reaction times [107], presumably stemming from the rapid and dynamic visual demands of such video games.

While this study has demonstrated an association between physical activity and mental health on the one hand, and video gaming and cognitive abilities on the other, it is important to acknowledge a number of clear limitations. First, there are inadequacies in the instruments used to measure our independent and dependent variables. Cognitive outcomes were derived from a comprehensive 12-task battery, able to provide insight on multiple domains of cognition, whereas mental

health outcomes were captured using brief psychological screeners (GAD-2 and PHQ-2) intended to identify individuals at risk of anxiety and depression. It is important to acknowledge that these two markers alone provide an incomplete picture of mental health, and that exercise and video-gaming may be associated with improvements in other aspects of mental well-being. In the same vein, it is possible that our measures of cognition were not sensitive to the benefits of exercise – although this seems less likely considering that the cognitive battery has been used in numerous other studies of human cognition. We also must acknowledge the limitations of using self-reported estimates of the frequency and duration of physical exercise and playing video games as it is well known that people over-report [51,108]. To overcome this, we categorized individuals based on their self-reported frequency of the activity; although the accuracy of individuals' self-report judgements may be low, the classification of individuals into categories (i.e., high or low extremes) based on their estimates is substantially better [37,52,53]. Obviously, objective data would produce more accurate measurements, even for classifying individuals [109], but within the present study obtaining gold standard accelerometer data was simply not feasible due to the international online design and costs associated with our large sample size. Personal activity monitors (e.g., FitBit, Apple Watch, etc.) provide an interesting potential alternative for acquiring objective physical activity data, but there are significant issues that first need to be overcome, such as addressing concerns related to data privacy, establishing the quality and reliability of data collected from dozens of different device types across multiple platforms with proprietary measurements, and the need for techniques to harmonize data from across these platforms [109]. Future work will investigate the feasibility of incorporating participants' existing personal physical activity tracking devices into an online study of this type. It is also worth noting the different time frames evaluated by the questionnaires for physical activity (PAAQ: hours of activity per week over the past week) and video gaming (VGQ; average hours of video gaming per week over the past year). This was simply a consequence of using established questionnaires, though we considered this discrepancy by asking participants if their PA estimate was representative of their usual routine, which most participants indicated was the case. Nonetheless, it could be that any cognitive benefits afforded by long-term physical exercise were not detected using this particular assessment; however, this seems unlikely given previous research that has demonstrated a positive relationship between acute exercise and cognition [12]. Despite the limitations noted here, the fact that we observed some significant associations for both cognitive and mental health, and exercise and video-gaming, suggests that the null results are not simply due to inadequacies of the instruments used, nor an insufficient sample size.

Another limitation is demographic bias. For example, our international sample was relatively skewed toward older adults, and roughly 70% of the sample exceeded the WHO exercise guidelines. This may potentially be due to recruitment methods (e.g., radio broadcasts, news articles, Facebook, etc.) that are likely not as popular among younger audiences. In any case, the sample is likely not representative of the general population, which may limit the generalizability of results to younger or less active populations, although the fact remains that among a large population of adults with a tendency towards regular exercise, a positive association was observed with mental health. Third, most studies of exercise, including ours, have investigated the chronic effects of regular exercise (e.g., WHO guidelines > 150 minutes/ week), rather than its acute (immediate) effects. We cannot, therefore, rule out the possibility that exercise exerts an immediate effect on cognition and several studies have suggested that this is the case [12,110,111]. Finally, a robust literature exists suggesting that action games exert a disproportionate effect on cognitive performance relative to non-action computer games [32]. We were neither able to confirm nor refute this association in our sample as there were insufficient numbers of action gamers to make a statistically meaningful comparison.

While we observed significant associations between physical exercise and mental health and between video gaming and cognitive performance, it is not possible to say that this is a causative relationship based on the current data. For example, more depressed or more anxious individuals may be less likely to exercise than their less depressed or less anxious counterparts. This explanation seems unlikely, however, because the association between physical activity and mental health scores was weakest among those with higher depression or anxiety ratings, suggesting that these symptoms were not the main drivers of the performance differences. Where video gaming and cognition is concerned, the

direction of the effect seems rather less open to question. That is to say, while it is theoretically possible that those who have better cognition are more likely to play video games regularly, it seems more probable that regular gaming exerts a direct effect on aspects of cognitive performance. In any case, the cross-sectional design of the present study does not allow us to draw any definitive conclusions surrounding causality or directionality. Future research may benefit from longitudinal follow-up to further investigate and determine the nature of this relationship over time.

These limitations aside, in the present study, we attempted to address some of the ambiguities arising from previous research though our methodological approach. For example, we accounted for variability within our cohort by adding levels of covariates to our analysis, as well as accounting for the influence of the *other* lifestyle factor (either video gaming or physical activity), which has frequently not been the case. That is, we investigated the effect of exercise *and* video gaming on mental health *and* cognitive ability, whereas many previous studies have primarily focused on one or the other of these factors. We also used a well-validated, comprehensive cognitive assessment tool which taps into a variety of cognitive functions and domains. Moreover, this was an international study (albeit with most participants coming from the U.K., Canada, and the U.S.A), rather than a group of participants tested within a laboratory setting and so the results may be more representative of the general population. That said, in at least one other study [112], with a large sample size (N = 945), no significant association was observed between exercise and cognition.

In conclusion, we have demonstrated that certain lifestyle interventions may enhance different aspects of cognitive and mental health, suggesting that one's overall brain health may benefit most from combinations of interventions. For example, activities that include a physical and cognitive component (e.g., mind-body exercise, 'exergaming', or yoga; [113–116]) may target multiple facets of brain health, potentially decreasing the likelihood of age-related cognitive impairment and protecting against mental health disorders. To build upon these findings, future studies should aim to incorporate an objective assessment of physical activity level (e.g., wearable devices), as well as more comprehensive mental health assessment tools. Furthermore, a longitudinal study design would allow for more definitive and generalizable conclusions regarding the nature of the relationships between exercise, video gaming, and brain health. Research in this domain is important to identify which aspects of brain health are affected by certain lifestyle factors, in order to support health policies and inform individuals to choose activities that will preserve their mental and cognitive health with age.

## Supporting information

**S1 Table. Descriptive Statistics of all Cognitive Tests.**
(DOCX)

**S2 Table. Results of Likelihood Ratio Tests for Lifestyle Factors Across all Brain Health Scores.**
(DOCX)

**S3 Table. Pairwise Differences in Marginal Predicted Probabilities of Mental Health Scores.** Pairwise differences in the marginal predicted probabilities of mental health scores between levels of physical activity (yes – no; i.e., met WHO guidelines vs did not meet WHO guidelines). p-values were Bonferroni corrected for N = 6 comparison for PHQ-2 and N = 4 for the GAD-2. SE = standard error, z.ratio = z-statistics, p.adj = corrected p-value.
(DOCX)

**S4 Table. Pairwise Tests of Differences in Marginal Means Between Gaming Subtypes.** Results for all pairwise tests of differences in the marginal means between gaming subtypes, within each cognitive score. p-values were Bonferroni corrected for N = 3 comparisons. SE = standard error, df = degrees of freedom, t.stat = t-statistic, p.adj = corrected p-value.
(DOCX)

**S5 Table. Results of Likelihood Ratio Tests for Cognitive Score Testing Interactions Between Age and Lifestyle Factors. df.** H1 = degrees of freedom for the full (H1) model, LL.H1 = Log Likelihood of the full model, df = degrees of

freedom difference between full and nested (H0) models (also the df of the Chisq test), LR = likelihood ratio, p.unc = uncorrected p-value, p.adj = corrected p-value.
(DOCX)

## Acknowledgments

We would like to thank Mike Battista and Ira Gupta for their assistance with study implementation and overall support for the project. Thank you to our colleagues at the Science Museum Group and to BrainsCAN at Western University for endorsing the study. A.M.O. is a Fellow of the CIFAR Brain, Mind, and Consciousness program.

## Author contributions

**Conceptualization:** Sydni G. Paleczny, Conor J. Wild, Roger Highfield, Adrian M. Owen.

**Data curation:** Conor J. Wild, Alex Xue.

**Formal analysis:** Conor J. Wild, Alex Xue.

**Funding acquisition:** Adrian M. Owen.

**Investigation:** Sydni G. Paleczny, Conor J. Wild.

**Methodology:** Conor J. Wild, Alex Xue.

**Project administration:** Sydni G. Paleczny, Roger Highfield.

**Resources:** Roger Highfield, Adrian M. Owen.

**Supervision:** Adrian M. Owen.

**Writing – original draft:** Sydni G. Paleczny, Conor J. Wild.

**Writing – review & editing:** Alex Xue, Adrian M. Owen.

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
