## [Decision Letter · Decision Letter 0]

18 May 2025

Dear Dr. Paleczny,

Thank you for submitting your manuscript to PLOS ONE. After careful consideration, we feel that it has merit but does not fully meet PLOS ONE’s publication criteria as it currently stands. Therefore, we invite you to submit a revised version of the manuscript that addresses the points raised during the review process.

**Please provide a point-by-point and detailed response to all the reviewers' comments.**

We look forward to receiving your revised manuscript.

Kind regards,

Dr. Mohammad Mofatteh, PhD, MPH, MSc, PGCert, BSc (Hons), MB BCh (c)

Academic Editor

PLOS ONE

Journal Requirements:

The cognitive tests used in this study are marketed by Creyos (formerly Cambridge Brain Sciences), of which A.M.O. is the chief scientific officer, C.J.W. is the director of data science, and S.G.P. is staff scientist. Under the terms of the existing licensing agreement, A.M.O. and his collaborators are free to use the platform at no cost for their scientific studies, and that such research projects neither contribute to, nor are influenced by, the activities of the company. Consequently, there is no overlap between the present study and the activities of Creyos, nor was there any cost to the authors, funding bodies, or participants who were involved in the study.

3. Please remove all personal information, ensure that the data shared are in accordance with participant consent, and re-upload a fully anonymized data set.

Additional Editor Comments :

Please provide a point-by-point and detailed response to all the reviewers' comments.

Reviewers' comments:

Reviewer's Responses to Questions

**Comments to the Author**

1. Is the manuscript technically sound, and do the data support the conclusions?

Reviewer #1: Partly

Reviewer #2: Yes

2. Has the statistical analysis been performed appropriately and rigorously?

Reviewer #1: Yes

Reviewer #2: Yes

3. Have the authors made all data underlying the findings in their manuscript fully available?

Reviewer #1: Yes

Reviewer #2: Yes

4. Is the manuscript presented in an intelligible fashion and written in standard English?

Reviewer #1: Yes

Reviewer #2: Yes

Reviewer #1: 1. Minor formatting issues should be addressed—for example, in the Methods section, some subsection headers appear at the bottom of a page with text continuing on the next, which affects readability.

2.In the Discussion, the authors justify using categorical self-report data over a continuous measure for classifying gaming frequency. A brief explanation of why this approach is preferable would improve clarity.

3.There are some limitations that should be adequately addressed:

a) Cross-sectional design: Causality cannot be established (e.g., do healthier individuals exercise more, or does exercise improve health?). The authors acknowledge this and suggest future longitudinal studies.

b) Self-report bias: Physical activity and gaming hours were self-reported. While categorization may reduce inaccuracies, objective measures (e.g., accelerometers) would strengthen the validity of the findings.

c) Mental health measures: The GAD-2 and PHQ-2 are brief screening tools, not comprehensive diagnostic instruments. As such, the study may overlook more nuanced mental health effects.

d) Sample demographics: The sample is skewed toward older adults (mean age 55) with high exercise adherence (73% met WHO guidelines), which may limit generalizability to younger or less active populations.

Reviewer #2: Dear Authors,

Thank you for this well-designed study.

The study design, evaluations, and analyses appear to have been executed with rigor and depth. The level of detail provided regarding the tests and measurements conducted is deemed adequate. To further enhance the quality of the manuscript, please refer to the recommendations outlined below:

- The objective statement, located at the end of the introduction section, should be refined to provide greater clarity and emphasize the significance and potential beneficial impacts of the study.

- In the conclusions section, please include concise, point-by-point recommendations directed at scientists, outlining specific adjustments to enhance the quality of future research on this or related topics.

Best regards

**Do you want your identity to be public for this peer review?** For information about this choice, including consent withdrawal, please see our Privacy Policy

Reviewer #1: **Yes: ** João Zambujal-Oliveira

Reviewer #2: No

---

## [Author Response · Author response to Decision Letter 1]

18 Jun 2025

Dear Dr. Mofatteh,

Thank you very much for your response, for the Reviewers’ thorough assessment of our manuscript, and for allowing us the opportunity to incorporate all necessary revisions. We greatly appreciate your consideration, and we have addressed all of the points raised below and in our Response Letter with our responses in blue font.

Overall, aside from formatting revisions, we noted that the Reviewers’ comments primarily focused on certain limitations related to our study design and measures. We agree with all these points, and within the manuscript we have acknowledged these limitations and how they might affect our results. We believe that these weaknesses, common to many such cross-sectional studies, are offset somewhat by the advantages of our design; specifically, the ability to collect a rich dataset (i.e., multiple standardized questionnaires and a comprehensive cognitive battery) from a large and diverse sample of the population. It is also worth noting that the Reviewers did not raise any concerns, statistical or otherwise, about how we analyzed this comprehensive dataset to test our hypotheses.

Regardless, we maintain that our study’s limitations do not discount or undermine the importance of our findings. Among a large sample of adults with a tendency toward regular exercise, we observed a positive relationship between self-reported exercise levels and mental health but not cognition, and a positive relationship between video gaming and cognition but not mental health. As we state in the manuscript: “Despite the limitations noted here, the fact that we observed some significant associations for both cognitive and mental health, and exercise and video-gaming, suggests that the null results are not simply due to inadequacies of the instruments used, nor an insufficient sample size.” Nevertheless, on the basis of the Reviewers’ feedback, we have elaborated on our methodological decisions to provide readers with additional clarification. We have also justified the use of certain measures as they relate to the present study design.

Thank you again for reconsidering our manuscript. Should anything remain unclear or if there are any additional points requiring our attention, please do not hesitate to contact us.

Sincerely,

Sydni Paleczny (spaleczn@uwo.ca) & Conor Wild (cwild@uwo.ca) on behalf of all co-authors.

1. Thank you for pointing us to the correct guidelines. All formatting has now been updated within the main manuscript, title page, and supporting information to adhere to these requirements.

2. We confirm that our declaration of competing interests does not alter our adherence to all PLOS ONE policies, and we have added the required statement to our Cover Letter declaration as follows:

“The cognitive tests used in this study are marketed by Creyos (formerly Cambridge Brain Sciences), of which A.M.O. is the chief scientific officer, C.J.W. is the director of data science, and S.G.P. is staff scientist. Under the terms of the existing licensing agreement, A.M.O. and his collaborators are free to use the platform at no cost for their scientific studies, and that such research projects neither contribute to, nor are influenced by, the activities of the company. Consequently, there is no overlap between the present study and the activities of Creyos, nor was there any cost to the authors, funding bodies, or participants who were involved in the study. This does not alter our adherence to PLOS ONE policies on sharing data and materials.”

Thank you for updating this within the online submission form on our behalf.

3. The complete raw dataset, including individual raw files and a final pre-processed dataset as described in the manuscript, are available at https://borealisdata.ca/dataset.xhtml?persistentId=doi:10.5683/SP3/WUYAGU. Participants consented for their fully anonymized data to be made available on this institutional dataverse repository. All identifying information has been removed from the individual data and preprocessed files.

4. Thank you for directing us to the corresponding formatting requirements. We have updated all in-text citations (e.g., “S1 Fig”) and added “Supporting information” as a Level 1 heading at the bottom of the manuscript which lists all file titles and labels.

Reviewer 1 #1. Thank you for bringing this to our attention. Originally, this issue may have been due to file type conversions – our apologies. We have updated the formatting accordingly so that all section and subsection headers appear directly above the following text. We have also updated all formatting requirements as per the PLOS guidelines.

Reviewer 1 #2. Thank you. We agree that further clarification of this method would be helpful for readers. We have therefore elaborated on why this categorical approach was used for classifying gaming frequency in the Methods section, on Page 7, Line 202:

“Previous studies have shown that, where self-report data is concerned, the categorization of individuals into extremes based on their reported quantity or frequency of the behaviour (e.g., whether or not they met activity guidelines, based on self-reported minutes) is more reliable than the continuous reported measure (36,53). For example, in their original publication of the PAAQ, Colley et al. show that despite weak correlations (best case Pearson r=0.36) between self-reported physical activity and objective measurements from accelerometers, the proportion of participants that met physical activity guidelines according to these methods were somewhat similar (39% vs. 46%) and these classifications were in agreement 67% of the time. Similarly, Bediou et al. (32) note that the video game questionnaire used in this study was designed only to distinguish frequent gamers from those who partake very infrequently.

Reviewer 1 #3.a) Thank you for raising these important points. We agree that this is a limitation that must be addressed, and have added additional detail to the following paragraph of our Discussion section (additions in bold). Page 31, Line 668:

“While we observed significant associations between physical exercise and mental health and between video gaming and cognitive performance, it is not possible to say that this is a causative relationship based on the current data. For example, more depressed or more anxious individuals may be less likely to exercise than their less depressed or less anxious counterparts. This explanation seems unlikely, however, because the association between physical activity and mental health scores was weakest among those with higher depression or anxiety ratings, suggesting that these symptoms were not the main drivers of the performance differences. Where video gaming and cognition is concerned, the direction of the effect seems rather less open to question. That is to say, while it is theoretically possible that those who have better cognition are more likely to play video games regularly, it seems more probable that regular gaming exerts a direct effect on aspects of cognitive performance. In any case, the cross-sectional design of the present study does not allow us to draw any definitive conclusions surrounding causality. Future research may benefit from longitudinal follow-up to further investigate and determine the nature of this relationship over time.”

Reviewer 1 #3.b) Thank you for this suggestion. We agree that it’s worth discussing the use of objective measures (and the challenges with this approach in relation to our methodological design) in order to build upon findings and improve future research. We have therefore added the following detail to Pages 29-30, Line 630:

“We also must acknowledge the limitations of using self-reported estimates of the frequency and duration of physical exercise and playing video games as it is well known that people over-report (51,101). To overcome this, we categorized individuals based on their self-reported frequency of the activity; although the accuracy of individuals’ self-report judgements may be low, the classification of individuals into categories (i.e., high or low extremes) based on their estimates is substantially better (36,52,53). Obviously, objective data would produce more accurate measurements, even for classifying individuals (102), but within the present study obtaining gold standard accelerometer data was simply not feasible due to the international online design and costs associated with our large sample size. Personal activity monitors (e.g., FitBit, Apple Watch, etc.) provide an interesting potential alternative for acquiring objective physical activity data, but there are significant issues that first need to be overcome, such as addressing concerns related to data privacy, establishing the quality and reliability of data collected from dozens of different device types across multiple platforms with proprietary measurements, and the need for techniques to harmonize data from across these platforms (102). Future work will investigate the feasibility of incorporating participants’ existing personal physical activity tracking devices into an online study of this type.

Reviewer 1 #3.c) Response from authors: Thank you for raising this important point. We have considered and acknowledged the limitations of using brief screening tools, although we feel they ultimately do not discount the importance of the present findings. For example, one could argue that the limited mental health screeners might produce null results, but we have instead shown that associations between (self-reported) physical activity and mental health can be detected with these simple two-item questionnaires. Furthermore, our decision to use these screeners was informed by our previous research (Wild et al. 2022) in which we demonstrated an association between mental health (assessed with these screeners), demographics, and long-term symptoms of COVID-19 infection.

Nevertheless, we fully agree that a more complete picture of mental health provided by more comprehensive instruments could reveal more nuanced effects related to mental health. For additional clarity, we have specified “GAD-2 and PHQ-2” within this section of the Discussion, Page 29, Line 621:

“there are inadequacies in the instruments used to measure our independent and dependent variables.” Specifically, “mental health outcomes were captured using brief psychological screeners (GAD-2 and PHQ-2) intended to identify individuals at risk of anxiety and depression. It is important to acknowledge that these two markers alone provide an incomplete picture of mental health, and that exercise and video-gaming may be associated with improvements in other aspects of mental well-being.”

Reviewer 1 #3.d) We agree with the noted limitations of our sample, and have acknowledged this in our Discussion section, Page 31 Line 657:

“Another limitation is that roughly 70% of our international sample exceeded the WHO exercise guidelines, which is likely not representative of the general population. The sample was also relatively skewed toward older adults, potentially due to recruitment methods (e.g., radio broadcasts, news articles, Facebook, etc.) that are likely not as popular among younger audiences. Nevertheless, the fact remains that among a large population of adults with a tendency towards regular exercise, a positive association was observed with mental health.”

We have added the following sentence for clarity. Page 30, Line 639: “This may limit the generalizability of results to younger or less active populations.” Ultimately, while we acknowledge this limitation, we again believe it does not discount our main finding: a statistically significant relationship between regular activity and mental health within our studied population of generally active adults.

Reviewer 2 #1. Thank you for this suggestion. To emphasize the significance and potential beneficial impacts of our study, we have added additional details to the final paragraph of the Introduction beginning on Page 3, Line 128:

“To address some of the outstanding questions in this field, we performed an international online study in a large sample of younger, middle-aged, and older adults, with the objective of better characterizing the relationships between physical activity, video game playing, cognition, and mental health. To measure various aspects of lifestyle and brain health, along with demographics and other health-related information, we deployed multiple validated questionnaires alongside a comprehensive cognitive test battery. The Creyos online cognitive testing platform, which includes twelve tasks that collectively assess a broad range of cognitive abilities, has been used in numerous large cohort studies (46–48) and provides detailed information about a variety of brain functions and the cognitive domains that they support. Based on the existing literature, we hypothesized that both exercise and video gaming would be related to cognition and/or mental health, although the exact nature of this relationship would likely vary in terms of the benefits observed – for example, the cognitive domains that showed associations with these lifestyle factors – and when during the lifespan they occur. To our knowledge, simultaneously assessing the effect of physical activity and video gaming on both cognitive performance and mental health in such a large sample has not been done previously. An advantage of this approach is that the profiles of brain health associated with each lifestyle factor can be directly compared because they are measured using the same instruments (i.e., questionnaires and cognitive tasks). The results from this largescale cross-sectional study will set the stage for future targeted studies of interventions and causal effects that will be of significant benefit for shaping health policy, informing individual lifestyle choices, and guiding future research on interventional approaches to brain health and longevity.

Reviewer 2 #2. We have added the following future directions in our conclusions section, Page 32 Line 708:

“To build upon these findings, future studies should aim to incorporate an objective assessment of physical activity level (e.g., wearable devices), as well as more comprehensive mental health assessment tools. Furthermore, a longitudinal study design would allow for more definitive and generalizable conclusions regarding the nature of the relationships between exercise, video gaming, and brain health.”

---

## [Decision Letter · Decision Letter 1]

17 Jul 2025

Dear Dr. Paleczny,

Thank you for submitting your manuscript to PLOS ONE. After careful consideration, we feel that it has merit but does not fully meet PLOS ONE’s publication criteria as it currently stands. Therefore, we invite you to submit a revised version of the manuscript that addresses the points raised during the review process.

**Please provide a detailed point-by-point response along with updated version of the manuscript.**

We look forward to receiving your revised manuscript.

Kind regards,

Mohammad Mofatteh, PhD, MPH, MSc, PGCert, BSc (Hons), MB BCh (c)

Academic Editor

PLOS ONE

Journal Requirements:

Additional Editor Comments:

Please provide a detailed point-by-point response along with updated version of the manuscript.

Reviewers' comments:

Reviewer's Responses to Questions

**Comments to the Author**

Reviewer #1: (No Response)

Reviewer #2: All comments have been addressed

2. Is the manuscript technically sound, and do the data support the conclusions?

Reviewer #1: Partly

Reviewer #2: Yes

3. Has the statistical analysis been performed appropriately and rigorously?

Reviewer #1: No

Reviewer #2: Yes

4. Have the authors made all data underlying the findings in their manuscript fully available?

Reviewer #1: Yes

Reviewer #2: Yes

5. Is the manuscript presented in an intelligible fashion and written in standard English?

Reviewer #1: Yes

Reviewer #2: Yes

Reviewer #1: -Provide justification for why Creyos is suitable for all age groups.

- Define "cognitive domains" with concrete examples.

- Acknowledge the limitations of self-reporting, which are noted in both the physical activity and gaming questionnaires; suggested solutions (e.g., categorical classification) should be clearly presented.

- Since recruitment was conducted online, specify what measures were taken—or could be taken—to mitigate bias toward individuals with higher internet literacy or an interest in cognition-related topics.

- The filtering criterion “Speaks >25 languages” requires a stronger justification.

- Some thresholds (e.g., “>20 units of alcohol/day”) seem arbitrary; provide a rationale or cite relevant literature.

- Consider removing repetition between text and tables (e.g., mental health results) to improve conciseness.

- Clarify how demographic bias may have led to an overrepresentation of active and older adults.

- Address any potential mismatch in measurement timing, such as physical activity being assessed weekly while gaming is assessed annually.

- Explain why the observational nature of the data limits conclusions about directionality.

- Suggest directions for future research, such as longitudinal study designs.

Reviewer #2: Dear Authors,

It seems that the suggested revisions were made successfully. With its current form, I believe that this manuscript can be a valuable addition to practical sport science applications.

Best regards,

**Do you want your identity to be public for this peer review?** For information about this choice, including consent withdrawal, please see our Privacy Policy

Reviewer #1: **Yes: ** João Zambujal-Oliveira

Reviewer #2: No

---

## [Author Response · Author response to Decision Letter 2]

11 Aug 2025

Reviewer #1:

- Provide justification for why Creyos is suitable for all age groups.

● Response from authors: We have added the following information (additions in bold) to Page 8 Lines 6-9:

○ “The 12-task Creyos battery assesses aspects of memory, attention, planning, and reasoning, using a web-based interface that can be self-administered at home through any internet browser (46). The Creyos battery has been used in similar large-scale studies to assess the effects of sleep (47), COVID-19 (48), and lifestyle factors on cognitive functioning among the general population (54) across the lifespan (55) in studies with children and adolescents (56,57) and older adults (54,58). Importantly, several studies have demonstrated that older adults with limited experience with technology can manage digital cognitive assessments, producing data that is as reliable as in-person assessment (59-61).

- Define "cognitive domains" with concrete examples.

● Response from authors: We have added the following information (additions in bold) to Pages 8 Line 23:

○ “These included three measures derived from a factor analysis (46,47) that reflect performance within three broad cognitive domains: 1) short-term memory (STM; the ability focus on and maintain task-relevant information in short-term memory), 2) reasoning (the ability to mentally transform information according to logical rules), and 3) verbal performance (the processing of information within the verbal or linguistic domain). These interpretations are derived from the characteristics shared by the tasks that load most heavily on each of the three factors observed by Hampshire et al. (2012), and are supported by neuroimaging results showing that distinct (and functionally consistent) brain networks are recruited by these combinations of tasks.”

- Acknowledge the limitations of self-reporting, which are noted in both the physical activity and gaming questionnaires; suggested solutions (e.g., categorical classification) should be clearly presented.

● Response from authors: The reviewer has requested the following changes:

(1) That we acknowledge the limitations of self-reporting

(2) That we suggest solutions (e.g., categorical classification)

We have already acknowledged the first point on Page 26 Line 16:

“We also must acknowledge the limitations of using self-reported estimates of the frequency and duration of physical exercise and playing video games as it is well known that people over-report (51,108).”

And, we offer three solutions to this problem: 1) categorization (our solution, given the data available), 2) accelerometers (not feasible for this kind of study), and 3) personal fitness trackers (which have their own issues that must be addressed in future work). These points are are all presented on Page 26 Line 19:

○ “To overcome this, we categorized individuals based on their self-reported frequency of the activity; although the accuracy of individuals’ self-report judgements may be low, the classification of individuals into categories (i.e., high or low extremes) based on their estimates is substantially better (37,52,53). Obviously, objective data would produce more accurate measurements, even for classifying individuals (109), but within the present study obtaining gold standard accelerometer data was simply not feasible due to the international online design and costs associated with our large sample size. Personal activity monitors (e.g., FitBit, Apple Watch, etc.) provide an interesting potential alternative for acquiring objective physical activity data, but there are significant issues that first need to be overcome, such as addressing concerns related to data privacy, establishing the quality and reliability of data collected from dozens of different device types across multiple platforms with proprietary measurements, and the need for techniques to harmonize data from across these platforms (109). Future work will investigate the feasibility of incorporating participants’ existing personal physical activity tracking devices into an online study of this type.”

- Since recruitment was conducted online, specify what measures were taken—or could be taken—to mitigate bias toward individuals with higher internet literacy or an interest in cognition-related topics.

● Response from authors: Indeed, bias may have affected the general representativeness of our sample, which we have mentioned in the Discussion (additions in bold) on Page 27 Line 19:

○ “Another limitation is demographic bias. For example, our international sample was relatively skewed toward older adults, and roughly 70% of the sample exceeded the WHO exercise guidelines. This may potentially be due to recruitment methods (e.g., radio broadcasts, news articles, Facebook, etc.) that are likely not as popular among younger audiences. In any case, the sample is likely not representative of the general population which may limit the generalizability of results to younger or less active populations”

- The filtering criterion “Speaks >25 languages” requires a stronger justification.

● Response from authors: Thank you for pointing this out. The correct filtering criterion should read “Speaks >7 languages” which was determined through visual inspection of histogram data to identify outliers. We have updated this criterion in Table 1, and added the following information (additions in bold) to Page 11 Lines 9-10:

○ “Those who reported unusually high numbers for numeric entry questions (according to visual inspection of histogram data to identify outliers), who had a cognitive score more than 4 standard deviations from the mean, or who were missing any data from the cognitive tests, mental health screeners, the PAAQ, or the VGQ were also excluded.”

- Some thresholds (e.g., “>20 units of alcohol/day”) seem arbitrary; provide a rationale or cite relevant literature.

● Response from authors: Thank you again for pointing this out. This threshold corresponds to the upper ~5% of the distribution and corresponds to discontinuities identified from visual inspection of histogram data. We have added detail (additions in bold) to our filtering criteria on Page 11 Lines 9-10:

○ “Those who reported unusually high numbers for numeric entry questions (according to visual inspection of histogram data to identify outliers), who had a cognitive score more than 4 standard deviations from the mean, or who were missing any data from the cognitive tests, mental health screeners, the PAAQ, or the VGQ were also excluded.”

- Consider removing repetition between text and tables (e.g., mental health results) to improve conciseness.

● Response from authors: Thank you for this suggestion. We have removed the aspects of the text that repeat data in tables for conciseness (Page 16).

- Clarify how demographic bias may have led to an overrepresentation of active and older adults.

● Response from authors: We have reworded and added detail (additions in bold) to Page 27 Line 19:

○ “Another limitation is demographic bias. For example, our international sample was relatively skewed toward older adults, and roughly 70% of the sample exceeded the WHO exercise guidelines. This may potentially be due to recruitment methods (e.g., radio broadcasts, news articles, Facebook, etc.) that are likely not as popular among younger audiences. In any case, the sample is likely not representative of the general population, which may limit the generalizability of results to younger or less active populations.”

- Address any potential mismatch in measurement timing, such as physical activity being assessed weekly while gaming is assessed annually.

● Response from authors: We have addressed the mismatch in measurement timing on Page 27 Line 7, and reworded for clarity (additions in bold):

○ “It is also worth noting the different time frames evaluated by the questionnaires for physical activity (PAAQ: hours of activity per week over the past week) and video gaming (VGQ; average hours of video gaming per week over the past year). This was simply a consequence of using established questionnaires, though we considered this discrepancy by asking participants if their PA estimate was representative of their usual routine, which most participants indicated was the case. Nonetheless, it could be that any cognitive benefits afforded by long-term physical exercise were not detected using this particular assessment; however, this seems unlikely given previous research that has demonstrated a positive relationship between acute exercise and cognition (12)."

- Explain why the observational nature of the data limits conclusions about directionality.

● Response from authors: We have acknowledged that our study design does not allow us to draw any conclusions surrounding causality, and added the following detail (additions in bold) to Page 28 Line 9:

○ “While we observed significant associations between physical exercise and mental health and between video gaming and cognitive performance, it is not possible to say that this is a causative relationship based on the current data. For example, more depressed or more anxious individuals may be less likely to exercise than their less depressed or less anxious counterparts. This explanation seems unlikely, however, because the association between physical activity and mental health scores was weakest among those with higher depression or anxiety ratings, suggesting that these symptoms were not the main drivers of the performance differences. Where video gaming and cognition is concerned, the direction of the effect seems rather less open to question. That is to say, while it is theoretically possible that those who have better cognition are more likely to play video games regularly, it seems more probable that regular gaming exerts a direct effect on aspects of cognitive performance. In any case, the cross-sectional design of the present study does not allow us to draw any definitive conclusions surrounding causality or directionality. Future research may benefit from longitudinal follow-up to further investigate and determine the nature of this relationship over time.”

- Suggest directions for future research, such as longitudinal study designs.

● Response from authors: Please see Page 29 Line 16 (relevant text highlighted):

○ “To build upon these findings, future studies should aim to incorporate an objective assessment of physical activity level (e.g., wearable devices), as well as more comprehensive mental health assessment tools. Furthermore, a longitudinal study design would allow for more definitive and generalizable conclusions regarding the nature of the relationships between exercise, video gaming, and brain health. Research in this domain is important to identify which aspects of brain health are affected by certain lifestyle factors, in order to support health policies and inform individuals to choose activities that will preserve their mental and cognitive health with age.”

Reviewer #2: Dear Authors,

It seems that the suggested revisions were made successfully. With its current form, I believe that this manuscript can be a valuable addition to practical sport science applications.

Best regards,

---

## [Decision Letter · Decision Letter 2]

5 Oct 2025

Characterizing the Cognitive and Mental Health Benefits of Exercise and Video Game Playing

PONE-D-25-17797R2

Dear Dr. Paleczny,

We’re pleased to inform you that your manuscript has been judged scientifically suitable for publication and will be formally accepted for publication once it meets all outstanding technical requirements.

Kind regards,

Egemen Mancı

Academic Editor

PLOS ONE

Additional Editor Comments (optional):

Reviewers' comments:

Reviewer's Responses to Questions

**Comments to the Author**

Reviewer #1: All comments have been addressed

Reviewer #2: All comments have been addressed

2. Is the manuscript technically sound, and do the data support the conclusions?

Reviewer #1: Yes

Reviewer #2: Yes

3. Has the statistical analysis been performed appropriately and rigorously?

Reviewer #1: Yes

Reviewer #2: Yes

4. Have the authors made all data underlying the findings in their manuscript fully available?

Reviewer #1: Yes

Reviewer #2: Yes

5. Is the manuscript presented in an intelligible fashion and written in standard English?

Reviewer #1: Yes

Reviewer #2: Yes

Reviewer #1: All main concerns appear to have been addressed. The revisions have strengthened the manuscript, which now meets expectations for methodological rigor, data transparency, and clarity of presentation.

Reviewer #2: Dear Authors,

It appears that the proposed changes were effectively implemented. In its present state, I am confident that this manuscript will significantly contribute to practical applications in sport science.

**Do you want your identity to be public for this peer review?** For information about this choice, including consent withdrawal, please see our Privacy Policy

Reviewer #1: **Yes: ** João Zambujal-Oliveira

Reviewer #2: No

---

## [Editor Report · Acceptance letter]

PONE-D-25-17797R2

PLOS ONE

Dear Dr. Paleczny,

I'm pleased to inform you that your manuscript has been deemed suitable for publication in PLOS ONE. Congratulations! Your manuscript is now being handed over to our production team.

Kind regards,

on behalf of

Dr. Egemen Mancı

Academic Editor

PLOS ONE